# Recent Pharmacological Options in Type 2 Diabetes and Synergic Mechanism in Cardiovascular Disease

**DOI:** 10.3390/ijms24021646

**Published:** 2023-01-13

**Authors:** Aikaterini Andreadi, Saverio Muscoli, Rojin Tajmir, Marco Meloni, Carolina Muscoli, Sara Ilari, Vincenzo Mollace, David Della Morte, Alfonso Bellia, Nicola Di Daniele, Manfredi Tesauro, Davide Lauro

**Affiliations:** 1Department of Systems Medicine, Section of Endocrinology and Metabolic Diseases, University of Rome Tor Vergata, 00133 Rome, Italy; 2Division of Endocrinology and Diabetology, Fondazione Policlinico Tor Vergata, 00133 Rome, Italy; 3Division of Cardiology, Fondazione Policlinico Tor Vergata, 00133 Rome, Italy; 4Department of Health Science, University of Magna Graecia, 88100 Catanzaro, Italy; 5Department of Systems Medicine, University of Rome Tor Vergata, 00133 Rome, Italy; 6Division of Internal Medicine—Hypertension, Department of Medical Sciences, Fondazione Policlinico “Tor Vergata”, 00133 Rome, Italy; 7Department of Neurology, Evelyn F. McKnight Brain Institute, University of Miami Miller School of Medicine, Miami, FL 33136, USA

**Keywords:** oxidative stress, insulin resistance, diabetes mellitus, cardiovascular risk, cardiovascular complications

## Abstract

Diabetes Mellitus is a multifactorial disease with a critical impact worldwide. During prediabetes, the presence of various inflammatory cytokines and oxidative stress will lead to the pathogenesis of type 2 diabetes. Furthermore, insulin resistance and chronic hyperglycemia will lead to micro- and macrovascular complications (cardiovascular disease, heart failure, hypertension, chronic kidney disease, and atherosclerosis). The development through the years of pharmacological options allowed us to reduce the persistence of chronic hyperglycemia and reduce diabetic complications. This review aims to highlight the specific mechanisms with which the new treatments for type 2 diabetes reduce oxidative stress and insulin resistance and improve cardiovascular outcomes.

## 1. Introduction

Long-term maintenance of physiological blood glucose levels is important in subjects with diabetes. One speaks of normal glucose homeostasis when the glucose level remains below 100 mg/dl after overnight fasting. The situation is different if the glucose values are between ≥100 mg/dL and <126 mg/dL or ≥126 mg/dL, in which case the person is considered to have prediabetes or diabetes [1]. It is estimated that approximately 460 million people worldwide suffer from type 2 diabetes mellitus (T2D). T2D is one of the most important non-communicable diseases (NCD), which, together with cancer, chronic obstructive pulmonary disease (COPD), and cardiovascular disease (CVD), are responsible for about 80% of all premature deaths [2,3,4].

The International Diabetes Federation estimates that by 2045 there will be over 700 million patients worldwide with DM. T2D is a complex and multifactorial disease in which, in addition to hyperglycemia, there are also defects in fat and protein metabolism with increased lipid levels and sarcopenia [5]. Persons with diabetes have a four- to five- times higher risk of developing cardiovascular disease compared to non-diabetics. Furthermore, morbidity and mortality in patients with T2D are mainly associated with chronic cardiovascular complications [6]. 

The majority of T2D patients have moderate or high levels of insulin resistance (IR) and the presence of several clinical chronic complications, which include CVDs, chronic kidney disease, retinopathy, liver dysfunction, and neuropathy, among others [7]. Despite significant advances in the treatment of T2DM, this disorder remains the second most important cause of a decline in global health-related life expectancy [8]. 

People with T2D have different levels of IR; this is associated with multiple risk factors such as obesity, dyslipidemia, hypertension, endothelial dysfunction, and procoagulant state [9]. Furthermore, the molecular basis of IR contributes directly to the pathogenesis of CVDs, independent of other metabolic defects [10]. IR is considered the “primum movens” of T2D and is associated with poor insulin action, decreased cell signaling, and decreased insulin sensitivity in peripheral target tissues (liver, muscle, and adipose tissue) where metabolic/nutrient utilization and storage decreases and catabolic processes increase [11]. Nowadays, IR is a common occurrence due to mitigating variables. Environmental factors, especially overweight and obesity, are believed to be the main causes [12]. Overeating and insufficient physical activity can result in hyperglycemia and hypertriglyceridemia, which can further compromise the functionality of pancreatic beta cells and exacerbate symptoms [13]. These variables contribute to a downhill spiral of increased IR, hyperinsulinemia, and raised blood glucose and free fatty acids; this compromises the function and metabolic state of pancreatic beta cells, leading to the development and progression of type 2 diabetes [14]. 

This review aims to show how new therapies for T2D modulate oxidative stress, improve insulin sensitivity, and lead to better cardiovascular outcomes.

## 2. Oxidative Stress

Hyperglycemia may impair glucose metabolism in the endothelium, leading to the formation of advanced glycosylated end products (AGE) or the polyol pathway [15]. AGE levels correspond to the degree of atherosclerotic disease in T2D and, in binding to the age receptor, they can activate redox transcription factors and inflammatory mediators, such as vascular cell adhesion molecules (RAGE) [16,17]. Adhesion molecules such as sICAM-1 (soluble intercellular adhesion molecule-1) and sVCAM-1 (soluble vascular cell adhesion molecule-1) impair endothelial function, which is a significant indication for the development of atherosclerosis and raises the risk of CVDs. Under the conditions of IR, chronically elevated blood levels of free fatty acids (FFA) reduce beta cell functionality and viability as well as lipotoxicity, which increases the risk of developing T2D [18]. 

Pancreatic beta cells are also exposed to islet amyloid peptide (IAPP) and proinflammatory cytokines such as interleukin 1-beta (IL -1b). Low-grade inflammation is considered part of the development of T2D. The islet cells of T2D contain a large number of immune cells as well as proinflammatory cytokines and chemokines [19]. 

Oxidative stress has been identified as one of the key mechanisms triggering IR and possibly leading to the pathophysiology of T2D [20]. Oxidative stress is characterized by an excess of oxidative species, such as reactive oxidative species (ROS) or reactive nitrogen species (RNS), which can cause cell damage or manipulate cell signaling pathways [9,21].

The overproduction or deficiency of ROS and RNS can alter cell homeostasis and cause various diseases. Today, numerous pathogenic mechanisms are believed to change the cell’s redox balance as the last common pathway [22]. The progressive increase in IR and subsequent damage to pancreatic beta cells and the onset of T2D is a pathophysiological process in which oxidative stress may play a crucial role. Indeed, cellular models of IR have continuously increased ROS levels [23]. The origin of this altered oxidant production is mainly mitochondria, and hyperglycemia may induce the alteration of mitochondrial morphology, including rapid fragmentation [24]. There are several lines of evidence that the overproduction of mitochondrial ROS (mainly superoxide) is related to the pathogenesis of T2D and its complications and that glucose can directly induce the overproduction of ROS [25]. 

Nevertheless, subsequent work has shown that high glucose levels stimulate several mitochondrial enzyme pathways, including activation of NADPH oxidase, uncoupling of NO synthase, and stimulation of xanthine oxidase [26]. However, other data suggest that high glucose concentrations suppress mitochondrial superoxide formation in pancreatic beta cells [27]. 

High concentrations of blood glucose increase glucose oxidation in cells, producing pyruvate and NADH, and these substrates are produced in abundance by mitochondrial complexes I, II, and III, increasing ROS. Mitochondrial dysfunction is associated with the occurrence of IR and various concomitant diseases, including CVDs [28]. 

Mitochondrial homeostasis is also modulated by mitochondrial dynamics such as mitochondrial fusion and fission [29]. Increased glucose, FFA, and leptin concentrations, along with the development of microvascular diseases such as neuropathy, retinopathy, and nephropathy, and macrovascular complications such as cardiac ischemia and stroke, are associated with ROS production [30]. 

Under these conditions, the antioxidant system and antioxidant enzymes are activated to reduce the production of ROS and prevent the formation of AGE or other harmful mediators of glucose metabolism or nuclear factor kappa-light-chain-enhancer of activated B cells (NF-kB) activation, thereby reducing oxidative levels and the chronic pro-inflammatory state [31]. Finally, hyperglycemia activates multiple metabolic signaling pathways leading to inflammation, cytokine release, cell death, and diabetic complications. Diabetes is a network of metabolic signaling, and science does not currently know how to control all cellular signaling pathways.

However, the production of ROS in the beta cells is necessary for regular glucose sensing and insulin secretion [32]. An imbalance between ROS and cellular antioxidants can lead to defects in glucose-stimulated insulin secretion (GSIS) and to the death and de-differentiation of pancreatic beta cells [33]. Oxidative stress and endoplasmic reticulum stress are thought to influence beta cell dysfunction with a lack of higher proinsulin secretion, which has been documented in T2D patients. Muscle contraction may trigger the production of ROS in skeletal muscle, but glucose uptake in response to exercise is expected in individuals with high IR and patients with T2D [34]. 

FFA increases oxidative stress in the beta cells. In particular, the long-chain FFA, which is not a substrate for the mitochondria, must be shortened by peroxisomal beta-oxidation and transported to the mitochondria to be broken down further [35]. Peroxisomal beta-oxidation produces H_2_O_2_ and thus increases oxidative stress. High glucose levels increase ROS production in beta cells by shifting their substrates to alternative metabolic pathways, such as the formation of dihydroxyacetone and diacylglycerol, as through diverse pathways, including autooxidation of glucose, hexosamine metabolism (glucosamine), the polyol pathway (sorbitol), advanced glycation end products (AGE) formation, and the activation of other enzymes (NADPH oxidase, etc.) [36]. In addition, high glucose content and FFA lead to increased superoxide formation via NADPH oxide [37]. Despite this, numerous research studies on the association between FFA and pancreatic beta cell dysfunction have generated contradictory results. These gaps may be partially explained by the possibility that the quality of FFA, and not just its quantity, modulates pancreatic beta cell function [38]. A recent study combining oral glucose tolerance assays with isoglycemic intravenous glucose tolerance testing revealed that a rise in FFA impairs the incretin-mediated enhancement of insulin production in both non-diabetic and T2D people [39]. 

## 3. Pharmacological Treatment

Several hypoglycemic agents may influence the mechanisms involved in the development of T2D. Metformin is one of the most frequently used medicines worldwide, and around 150 million individuals are treated with it [40]. Metformin also has a beneficial effect on the lipid metabolism of persons with diabetes-reducing CVDs [41]. Metformin-treated T2D patients reported lower mitochondrial ROS production and increasing concentrations of antioxidative enzymes, including Gpx1 and SIRT3, in addition to decreased leukocyte-endothelium interactions and reduced levels of ICAM-1 and P-selectin [42]. 

Several research papers demonstrate a reduction of oxidative stress biomarkers (Table 1) in diabetic patients and animals in response to metformin [43]. In animal studies, metformin was reported to reverse oxidative stress by activating the Sirt3-dependent pathway in response to arsenic-induced diabetes [44]. The primary effect in modulating oxidative stress is the inhibition of mitochondrial complex I (NADH: ubiquinone oxidoreductase) reducing ROS production [45]. In addition, metformin can inhibit apoptosis induced by high concentrations of FFA by modulating the response to oxidative stress and reducing proapoptotic protein kinase RNA-like endoplasmic reticulum kinase (PERK)/C/EBP homologous protein (CHOP) signaling [46]. Metformin can restore the action of PON1 (paroxonase 1), an antioxidant enzyme of the HDL lipoprotein that hydrolyses lipid peroxides in lipoproteins, especially in the LDL particle [47]. Thiazolidinediones, such as pioglitazone, appear to have direct antiatherosclerotic and antioxidant effects on T2D [48]. The antioxidant activity could be linked to Peroxisome proliferator-activated receptor γ (PPARγ) mediated increase in the activity of the Nrf2 (nuclear factor erythroid 2-related factor 2) transcription factor and its target genes [49]. In addition, PPAR activation also exerts an anti-inflammatory function by reducing the action of inflammatory transcription factors, such as nuclear factor-B, which may occur indirectly via elevated plasma adiponectin concentrations and activated AMPK [50]. This anti-inflammatory impact may explain the reduced fibrosis and fibrotic markers (TIMP1 and TGF-1) in animals following therapy with pioglitazone [51] (Table 1).

Moreover, the IRIS (Insulin Resistance Intervention after Stroke) trial showed that pioglitazone lowered the risk of stroke or heart attack in non-diabetic patients with insulin resistance and a recent stroke or transient ischemic attack. Additionally, pioglitazone was related to a decreased incidence of diabetes but an increased risk of weight gain, edema, and bone fractures [52]. The PROactive trial (PROspective pioglitAzone Clinical Trial In macroVascular Events) investigated the efficacy of pioglitazone in preventing macrovascular events in 5238 T2D patients. Pioglitazone reduces mortality from all causes, including non-fatal myocardial infarction and stroke [53].

The balance between the potential benefit of glitazones in reducing macrovascular events and the observed increase in the incidence of HF is of great concern. Consequently, the role of glitazones in treating T2D is not yet fully understood [54].

However, it remains to be elucidated how this antioxidant effect is modulated independently of the metabolic and hypoglycemic effects. Despite the promising results of these therapies, their use in CV risk reduction is limited by clinical evidence. New drugs recently used in T2D have shown promising developments in reducing CVD. 

### 3.1. Dipeptydilpeptidase-IV Inhibitor (DPP-IVi)

Glucagon-like peptide-1 (GLP-1) is an incretin hormone used as a newer therapy to lower fasting blood glucose levels in T2DM. However, GLP-1 is degraded by dipeptydilpeptidase-4 (DPP-IV), which is expressed by the brush border cells of the intestine [55]. DPP-IV is a transmembrane glycoprotein that is expressed in many tissues and is also present in soluble form in plasma. This protease can cleave incretin hormones and lead to their inactivation. DPP-IV plays a proinflammatory role, and its activity increases in plasma in diabetes and obesity [56]. Production of proinflammatory cytokines in lipopolysaccharide-treated macrophages and expression of inducible nitric oxide synthase (iNOS) increases in the presence of soluble DPP-IV (Table 1). An increase in DPP-IV blood levels has been associated with the production of ROS, innate immunity, and the activation of gene expression of receptors for advanced glycation end products [55]. DPP-IV inhibitors or gliptins, a new family of hypoglycemic medicines, increase the production of GLP-1 by inhibiting its deactivation and enhancing its action. As a result, glucagon release is reduced, and more glucose-induced insulin is secreted by islet cells [57]. DPP-IV inhibition possibly leads to the formation of Akt/endothelial NO synthase (eNOS), resulting in increased formation of NO and enhancement of fibroblast growth factor 2 (FGF-2)/early growth response protein 1 (EGR-1)/vascular endothelial growth factor A (VEGF-A) signaling [58]. Increasing NO synthesis leads to improved vascular smooth muscle function and a reduction of blood pressure [59]. In addition, inhibition of DPP-IV can prevent the inactivation of HMGB1 (high mobility group box 1), reduce expression of NOX -4, inhibit the JAK/STAT pathway (janus kinase/signal transducer and activator of transcription) and lead to restoration of the intracellular antioxidant glutathione and ATP levels [58].

Recent research has shown that antioxidants can reduce the complications of diabetes by decreasing the production of free radicals. Some natural compounds that act as DPP-IV inhibitors, such as alkaloids, steroids, phenolic acids, flavonoids, peptides, and peptidoglycans, also exhibit antioxidant properties.

Several studies, both in animal models and humans, suggest that these bioactive chemicals may be a potential treatment for T2DM in over 80% of the population [55,60].

### 3.2. Glucagon-like-1 Receptor Agonists (GLP-1RAs)

Glucagon-like-1 receptor agonists (GLP-1RAs) are a class of drugs that are resistant to the degrading effects of DPP-IV. GLP-1RAs increase glucose-induced insulin secretion, improve pancreatic beta cell function, suppress glucagon secretion under hyper- or euglycemic conditions, and reduce gastric emptying with a reduction of significant glycemic rises after a meal [61].

Increased effects on overnight and fasting plasma glucose and HbA1c (glycosylated hemoglobin) blood levels have been reported for long-acting GLP-1RAs. GLP-1RAs on background therapy with metformin and in combination with basal insulin showed additional effects on weight reduction (Table 1) (between 0.7 kg and 4.3 kg) and no risk of hypoglycemic episodes. The reduction of body weight was also observed in non-diabetic patients on GLP-1RAs [62].

In addition, GLP-1RAs can lower various cardiovascular risk factors such as systolic blood pressure (between 0.8 and 2.6 mmHg), plasma LDL cholesterol, and triglycerides, with a small increase in heart rate [63].

Since 2016, several CVDs studies have shown that GLP-1 RAs are effective in preventing cardiovascular events, such as acute myocardial infarction or stroke, as well as associated mortality and, in some cases, all-cause mortality [64,65,66]. GLP-1RAs may specifically regulate CV outcomes, reducing endothelial dysfunction and oxidative stress [67]. GLP-RAs may modulate the progression of atherosclerosis through anti-inflammatory effects on endothelial cells and reducing smooth muscle cell proliferation and pro-inflammatory effects of macrophages [68]. A meta-analysis of randomized controlled trials (RCTs) comparing GLP-1RAs to other diabetic treatments or placebo was associated with a significant decrease in inflammatory markers such as tumor necrosis factor-alpha (TNF-a) and oxidative stress biomarkers such as malondialdehyde (MDA), along with a substantial increase in blood adiponectin concentrations. This study represents a critical synthesis to better understand the various effects of GLP-1RAs that may impact the clinical course of T2D and CVDs [69]. The antioxidant effect of GLP-1RAs is mediated by receptor binding, activation of the cyclic adenosine monophosphate (cAMP), phosphatidylinositol-3 kinase (PI3K), and protein kinase C (PKC) signaling pathways, and activation of nuclear factor-erythroid 2 related factor 2 (Nrf-2). Increasing these cell-signaling pathways reduces oxidative stress biomarkers, which are elevated in response to various stress and metabolic factors (overweight or obesity, reduced physical activity, psychosocial stress, etc.) [70].

Nrf2 enhances the activity of islet cells under various situations, improves diabetes and obesity in mice, and enhances glucose utilization in skeletal muscle. In addition, data indicate that activating Nrf2 with GLP-1 and subsequently reducing oxidation levels can ameliorate DM metabolic abnormalities [71]. Furthermore, administration of GLP-1 (0.4 pmol/kg/min—2 h) reduced blood concentrations of 8-iso-PGF2 and nitrotyrosine in T1D and T2D patients [72]. Two-month administration of liraglutide (1.2 mg/die) to T2D patients decreased serum lipid hydroperoxides and haem oxygenase 1 [73].

A 12-month treatment with exenatide (10 mg/d) reduced postprandial glycemia and lipidaemia, which were directly correlated with a decrease in malondialdehyde and ox-LDL concentrations [74].

In vitro and in vivo studies have shown that the antioxidant effect of GLP-1RA may protect against chronic diabetic complications, particularly CVDs. Indeed, low GLP-1 levels and high oxidative stress biomarkers such as nitrotyrosine have been associated with cardiac remodeling and CVDs in patients with T2D [75]. In addition, liraglutide protects against oxidative stress by attenuating the S-glutathionylation of endothelial nitric oxide synthase (eNOS), increasing the bioavailability of NO and preserving endothelial function [76].

The transformation of macrophages into foam cells is the hallmark of plaque development and atherogenesis. In this contest, GLP-1RAs can suppress foam cell formation by activating the GLP-1R cell signaling pathway, reducing cholesterol esterification, and inhibiting cholesterol acyltransferase 1 expression [77]. In addition, GLP-1RAs induce autophagy and reduce activation of the PKA/CD36 pathway, thereby reducing ox-LDL uptake [78,79]. Furthermore, GLP-1RAs can protect cell functionality from oxidative stress damage by modulating AMPK (sterol regulatory element binding transcription factor-1 (SREBP1)) [80]. High ROS levels can accelerate the aging process and promote the development of CVDs. GLP-1RAs can partially inhibit this process by increasing adiponectin levels and inhibiting vascular matrix metalloproteinase 9 (MMP-9) and matrix metallopeptidase 2 (MMP-2) in APO E-/- mice [81].

GLP-1RAs can modulate the increased autophagy induced by high concentrations of ROS and pro-inflammatory molecules by upregulating HDAC6 (histone deacetylase 6) via stimulation of the GLP-1R-ERK1/2 pathway [82]. GLP-1RAs may also reduce ER stress by promoting protein folding via endoplasmic reticulum oxidoreductase (ERO1α) through AMPK-dependent activation, as well as by mitochondrial fusion [83]. These findings imply that GLP-1RAs may be essential in controlling oxidative stress in T2D patients, depending on their metabolic dysfunction. This may affect the evolution of T2D and its chronic consequences.

### 3.3. Sodium-Glucose Cotransporter-2 Inhibitors (SGLT-2i)

The more recent class of drugs approved as blood glucose-lowering agents for the treatment of T2D are the sodium-glucose cotransporter-2 inhibitors (SGLT-2i), also called gliflozines. SGLT-2i reduces the activity of the SGLT-2 cotransporter of sodium and glucose, located mainly in the brush border of the proximal renal tubules (S1 and S2 segments), which transports glucose via a sodium-dependent mechanism. Glucose is transported across the luminal membrane and into the extracellular space by glucose transporter 2 (GLUT-2), resulting in increased glucose excretion in the urine. Inhibition of SGLT2 decreases the reabsorption of filtered glucose at the proximal tubule, resulting in glycosuria and lowering blood glucose levels via an insulin-independent mechanism. The beneficial effects of SGLT-2i result from reduced gluconeogenesis and improved insulin sensitivity; however, they are associated with a higher glucagon response and insulin release from pancreatic beta cells [59,84] (Table 1).

Treatment of T2D patients with SGLT-2i also results in an average reduction of body weight of 1 to 3 kg, a reduction of HbA1c of 1%—13 mmol/mol, an average reduction in systolic blood pressure of 3 to 6 mmHg and a reduction of diastolic blood pressure of 0 to 2 mmHg [85,86,87,88]. Weight reduction with SGLT-2i is not only due to a reduction of body fluid but also to a significant loss of adipose tissue [89]. SGLT-2i have been identified as potent antioxidant drugs that can protect against oxidative damage by reducing free radical formation or activating the antioxidant system, as reported in animal model studies [90].

Indeed, it was found that in diabetic db/db mice (leptin receptor knockout mice) exposed to a high-fat diet and treated with SGLT-2i, redox status improved, and oxidative damage was reduced [91]. SGLT-2i can reduce the production of free radicals through direct and indirect mechanisms such as reducing hyperglycemia [92]. Dapagliflozin has been described to reduce the production of ROS by suppressing the expression of the enzyme Nox4 and improving hemodynamic status. In addition, nephrogenic ROS production was reduced by inhibiting Nox4, leading to the prevention of diabetic nephropathy [93]. Empagliflozin has a renal protective effect mediated by the reduction of oxidative stress related to the glucose-lowering effect and by the reduced expression of Nox1 and Nox2 [94]. In animal studies, SGLT-2i can reduce oxidative stress by modulating the production of pro-oxidant enzymes such as Nox, eNOS, and xanthine oxidase [95]. SGLT-2i reduced mitochondrial defects by improving the redox state; these data were obtained mainly in the brain. Empagliflozin inhibited the production of ROS and improved oxidative stress [96]. This effect modulated vascular dysfunction in the aortic vessels of diabetic rats by improving mitochondrial function and suppressing the activity of pro-oxidant substances [97].

Clinical studies of oxidative stress biomarkers found a significant reduction in T2D patients in response to SGLT-2i. In particular, four studies (3 randomized, one observational, n = 146 T2D patients) investigated 8-iso-PGF2a levels in three measured urine values and one serum value [98,99,100]. Nishimura et al. reported a substantial decrease in fasting urine 8-iso-PGF2a following 28 days of treatment with empagliflozin 10 mg/day (45.5%) and empagliflozin 25 mg/day (50.5%) versus placebo, in 60 T2D patients. [99] Similarly, the observational study by Solini et al. (n = 16 T2D patients) reported a significant decrease (30.3%) in urinary 8-iso-PGF2a levels two days after a single dose of 10 mg dapagliflozin [100].

Although these data are preliminary and, to our knowledge, no specific clinical trial has been designed to assess oxidative stress as a primary outcome, there are several indications that SGLT-2i can reduce oxidative stress associated with T2DM, followed by an improvement in insulin sensitivity and a reduction in the proinflammatory state associated with T2DM at short- and long-term follow-up [10]. These findings could explain, at least in part, the cardiovascular and renal advantages of SGLT-2i.

## 4. Discussion

In recent years, several RCTs have highlighted the beneficial effects of new pharmacological T2M options in improving cardiovascular outcomes.

In the saxagliptin assessment of vascular outcomes recorded in patients with diabetes mellitus-thrombolysis in the myocardial infarction 53 (SAVOUR-TIMI 53) study, individuals were enrolled with a history or high risk of CVE; additional treatment of saxagliptin compared to placebo seemed to have no impact on the primary endpoints (including non-fatal MI or ischaemic stroke and risk of CV death) (Table 2). However, saxagliptin was related to a considerably increased rate of heart failure hospitalizations. These findings led to several study designs investigating the effect of DPP-4 inhibitors on HF risk in T2DM patients [101]. Since the results of other clinical trials with DPP-4 inhibitors did not yield similar results to SAVOUR-TIMI 53, the increase in HHF observed in this study could be due to the study design or saxagliptin could have molecular properties that affect cardiomyocytes [58]. The cardiovascular outcome study of linagliptin (CAROLINA) was designed for T2D patients at high cardiovascular risk (Table 2). Patients were randomized to receive either 1 mg/day of glimepiride or 5 mg/day of linagliptin. Serious events such as HFH, MI, stroke, cardiovascular death, and death from all causes did not differ between the two groups, according to the results [102]. The CAROLINA proved that glimepiride and linagliptin are equivalent in terms of cardiovascular safety. Nevertheless, it was concluded that glimepiride should be restricted to patients with acute coronary syndrome and stroke, recent HF, and coronary events, as the population in this study was not sufficiently representative [102]. In contrast to SAVOUR-TIMI 53, the evaluating cardiovascular outcomes with sitagliptin (TECOS) and the cardiovascular and renal microvascular outcome study with linagliptin (CARMELINA) investigators concluded no difference in the incidence of HFH between the DPP-4 inhibitor and placebo groups [103] (Table 2).

The results of the EXAMINE (examination of cardiovascular outcomes with alogliptin versus standard of care) investigation showed that alogliptin is not related to an increase in CV events compared to placebo in patients with T2DM with acute coronary syndrome. In addition, contrary to the results of the SAVOUR-TIMI 53 trials, the EXAMINE study indicated no increase in the risk of new HFH but a modest increase in the rate of hospitalization in patients without a history of HF [104].

In contrast, long-acting GLP-1RAs, such as liraglutide, semaglutide, dulaglutide, and albiglutide, demonstrated favorable 3-point MACE results. Furthermore, the liraglutide effect and action in diabetes: evaluation of cardiovascular outcome results (LEADER) showed a decrease in primary outcomes linked to CV fatalities and death from all causes versus placebo. However, the rates of non-fatal MI, non-fatal stroke, and HFH did not differ significantly from the placebo. Moreover, liraglutide was associated with a decreased incidence of nephropathy. Nevertheless, this drug was associated with digestive side effects [105]. The study for the evaluation of lixisenatide in acute coronary syndrome (ELIXA) was the first CVOT of the GLP-1Ra class to be completed. ELIXA indicated that lixisenatide is non-inferior to placebo in terms of cardiovascular mortality, non-fatal stroke, and non-fatal myocardial infarction [106] (Table 2).

The exenatide study of cardiovascular event lowering (EXSCEL), which examined 14,752 participants for 3.2 years, found no significant difference between exenatide and placebo in MACE [107] (Table 2).

The semaglutide group demonstrated a substantial reduction in the primary endpoint (first occurrence of CV death, non-fatal MI, or non-fatal stroke) in the semaglutide and cardiovascular outcomes in patients with type 2 diabetes (SUSTAIN -6) study. However, the two groups did not significantly differ in overall mortality and hospitalization rates [108] (Table 2).

The cardiovascular events with a weekly incretin in diabetes study REWIND investigated once-weekly subcutaneous administration of dulaglutide over 5.4 years. The results showed a significant reduction in CV deaths, non-fatal MI, and non-fatal strokes. However, they showed no significant difference compared to placebo in death from all causes. The authors concluded that of 68.5% of patients with CV risk factors at baseline, only 31.5% had established cardiovascular disease. Therefore, these agents should be considered for glycemic control in people with previous CVD or CV risk factors [109] (Table 2).

Patients treated with Albiglutide were superior to placebo in CV mortality, non-fatal MI, and non-fatal strokes in the HARMONY study of 9463 participants. However, no change in all-cause mortality, cardiovascular death, or stroke was observed compared to the placebo [108].

Terzipatide is an investigational new drug for the treatment of T2D and obesity. This agent acts as a glucose-dependent insulinotropic polypeptide (GIP) and GLP-1 receptor agonist. Terzipatide may improve glycemic control by increasing insulin secretion from the pancreatic β-cell, increasing insulin sensitivity and reducing glucose-dependent glucagon levels via GLP-1Ras activity. In addition, Terzipatide has been associated with improvements in blood pressure, lipid profiles, and inflammatory biomarkers [110].

The cardioprotective effects of tirzepatide are currently being investigated in several ongoing studies as SURPASS-CVOT [111].

Empagliflozin was the first SGLT2i with CV results. The EMPAREG trial enrolled 7010 patients with T2D and proven ASCVD. The primary composite endpoint was MACE, which was reduced by 14% in the empagliflozin group. In the empagliflozin group, the relative risk of HFH was reduced by 35%, the risk of cardiovascular disease decreased by 38%, and the risk of death from all causes decreased by 32%. No significant differences were found for MI or stroke [112].

The DECLARE-TIMI Study examined dapagliflozin in T2DM patients with or without established ASCVD. The primary endpoints of cardiovascular death, nonfatal MI, and nonfatal stroke were not significantly reduced between the dapagliflozin and placebo groups. It was the first study to evaluate HFH as a primary endpoint, and the results suggested a substantial decrease [113] (Table 2). The aim of the CANVAS study was to examine the use of canagliflozin in persons with diabetes with and without established ASCVD. The canagliflozin group reported a 14% decrease in the key composite outcome, mostly attributable to a decrease in HFH. The canagliflozin group showed no superiority in death from all causes [112]. The CREDENCE Study investigated canagliflozin in diabetic patients with chronic kidney disease with or without CVD. The study CREDENCE showed that canagliflozin was superior to placebo in improving glycemic control and reducing adverse renal events [114].

In the VERTIS CV study, ertugliflozin did not achieve superiority in reducing major CV or secondary composite renal events. However, HFH was significantly reduced compared to the placebo. When used with standard therapy, ertugliflozin may reduce the risk of a sustained 40% drop in eGFR in individuals with DMT2 and established ASCVD [115].

The SCORED and SOLOIST-WHF trial were stopped prematurely due to loss of funding. However, both studies showed positive results on the revised endpoints. During the SCORED study, the primary endpoint decreased significantly, primarily due to a 33% reduction in HFH and urgent visits for heart failure (HR, 0.67; 95% CI, 0.55–0.82). The CV reduction in mortality from sotagliflozin was reported to be not significantly different from the placebo group [113].

The study SOLOIST-WHF investigated the effect of sotagliflozin in T2D patients with HFrEF. Sotagliflozin was neutral regarding CV mortality; however, it showed a 30% reduction in HFH and urgent visits for HF [116].

## 5. Conclusions

In recent years, T2D has been one of the major causes of the increase in CVD mortality (Figure 1). In T2D patients, the use of SGLT2i, GLP1-Ras, and DDP-IVi improves CVOTs and controls metabolic consequences. Several studies have shown that the use of novel antiglycemic drugs interrupts or attenuates several processes involved in the development of atherosclerosis, leading to a decrease in cardiovascular complications. T2D activates a number of pathogenic molecular processes that promote endothelial dysfunction, atherosclerosis, and cardiovascular events. T2D results in increased oxidative stress and inflammation, decreased availability of NO, altered vascular tone, increased permeability, and procoagulant condition, all of which contribute to the formation, development, and degeneration of atherosclerotic plaques. The main goals of treatment are to improve quality of life, reduce hospitalization, and decrease mortality.

## Figures and Tables

**Figure 1 ijms-24-01646-f001:**
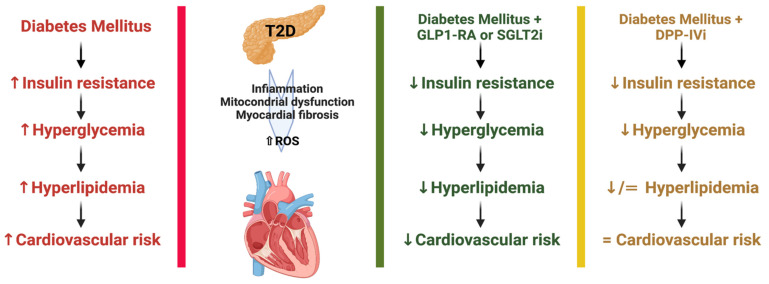
Recent pharmacological options for Type 2 Diabetes and efficacy at cardiovascular risk: from the data presented, we can affirm that GLP1-RA and SGLT2i have demonstrated, regarding the duration of the studies, the inclusion of subjects with diabetes, and the primary outcome vs. control group demonstrated more efficacy to reduce cardiovascular risk, (↑: increased, ↓: decreased).

**Table 1 ijms-24-01646-t001:** Summary of the effects of antidiabetic drugs (⇑: increased, ⇓: decreased): in this table, there are summarized the principal effects of the antiglycemic drugs such as metformin, thiazolidinediones, DPP-IVi, GLP1—Ras, and SGLT2i.

Pharmacological Treatment				
Metformin	Thiazolidinediones	DPP-IVi	GLP1-RAs	SGLT2i
⇓risk of hypoglycemia ⇓hyperlipidemia ⇓lower oxidative stress	⇑insulin sensitivity Control of blood pressure and hyperlipidemia ⇓infiammation ⇓cardiovascular events	⇓glycemic variability Glycemic control ⇓oxidative stress	⇓reduced glucose concentration ⇓oxidative stress and inflammation ⇓cardiovascular and adverse events Body weight controll	Improve glycemic control ⇓oxidative stress and inflammation ⇓hyperlipidemia

**Table 2 ijms-24-01646-t002:** Recent pharmacological options and Cardiovascular Outcome Trials: data regarding the duration of the studies, the inclusion of subjects with diabetes, and the primary outcome vs. control group (↑: increased, ↓: decreased).

Class	Trial	Duration of the Study (In Years)	Diabetes	Primary Outcome in Drug vs. Control (%)
DPP-IVi	SAVOR/TIMI 53 (saxagliptin)	2.1	Yes	↑ 7.3% vs. 7.2%
	CARMELINA (Linagliptin)	2.2	Yes	↑ 12.4% vs. 12.1%
	CAROLINA (linagliptin)	6.3	Yes	↑ 11.8% vs. 12%
	TECOS (sitagliptin)	3	Yes	↓ 11.4% vs. 11.6%
	EXAMINE (alogliptin)	1.5	Yes	↓ 11.3% vs. 11.8%
SGLT2i	EMPEROR-reduced (empagliflozin)	1.4	With/without	↓ 15.8% vs. 21.0%
	EMPA-REG (empagliflozin)	3.1	Yes	↓ 37.4% vs. 43.9%
	Emperor-presrved (empagliflozin)	2.4	With/without	↓ 13.8% vs. 17.1%
	Declare-TIMI (dapagliflozin)	4.2	Yes	↓ 22.6% vs. 24.2%
	DAPA-HF (dapagliflozin)	1.7	With/without	↓ 11.6% vs. 15.6%
	CANVAS (canagliflozin)	3.6	Yes	↓ 26.9% vs. 31.5%
	CREDENCE (canagliflozin)	2.6	Yes	↓ 11.1% vs. 15.5%
	VERTIS CV (ertuglifozin)	3.5	Yes	n 11.9% vs. 11.9%
	SOLOIST-WHF (sotagliflozin)	0.9	Yes	↓ 51.0% vs. 76.3%
	SCORED (sotagliflozin)	1.3	Yes	↓ 56% vs. 75%
GLP1-RAs	LEADER (liraglutide)	3.8	Yes	↓ 13% vs. 14.9%
	REWIND (dulaglutide)	5.4	Yes	↓ 12% vs. 13.4%
	HARMONY (albiglutide)	1.6	Yes	↓ 7% vs. 9%
	SUSTAIN-6 (semaglutide)	2.1	Yes	↓ 6.6% vs. 8.9%
	ELIXA (lixisenatide)	2.1	Yes	↓ 13.4% vs. 13.2%
	EXSCEL (exenatide)	3.2	Yes	↓ 11.4% vs. 12.2%
GIP/GLP-1Ra	SUPRASS (terzipatide)	ongoing	Yes	ongoing

## Data Availability

Not applicable.

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
