# Peer review of "Recent Pharmacological Options in Type 2 Diabetes and Synergic Mechanism in Cardiovascular Disease"

_ijms, 2023, doi:10.3390/ijms24021646_

Round 1

Reviewer 1 Report

Relevant and thorough review

Author Response

We thank you for your comment, and we appreciate your comment regarding our paper.

Reviewer 2 Report

Overall informative article. The following changes are recommended. 

1. Discern animal vs human studies when discussing specific data

2. Do not use "diabetic" use persons/subjects with diabetes 

3. Correct typos throughout the text

4. Many acronyms used that are not defined, please define

5. Multiple words used for agents that lower glucose -( antiglycemic, hypoglycemic, antidiabetic) pick one and use throughout the text, would avoid antidiabetic

6. Add information on Pioglitazone and stroke IRIS and PRoactive 

7. in Table 1- DDP4 section Correct the symbol for arrow (decrease/increase) and remove " ameliorate endothalial function" or use a legend to explain this 

8. In Table 1 SGLT2 section- first statement " decrease glycemic control" correct to "improve glycemic control "

9. Section 3.1 - line 1 - GLP1  is not a new therapy - use "newer" - The FDA approved the first GLP-1 receptor agonist exenatide in 2005

10. Last sentence in section 3.1 - add if animal or human studies?

11. Section 3.3 first sentence,  SGLT2 drugs are the most recent class ...Since the new GIP/GLP  drug is out and approved you may want to say "the more recent class"

12. the New GIP/GLP combination drug should be included in this review. 

in section 3, You may want to consider creating incretins based class subheading to include DDP4, GLP and GIP and SGLT2 as a second subheading 

or just add GIP/GLP drug in a new subheading in section 3.

13. Should add drug names next to trial names in table 2

14. I would change the Figure 1 to remove candy wrapping figure for T2DM as that may overemphasize the association of food/candy and diabetes risk.  Use simple nonfood figure. 

15. update the references for ADA standards of care to most current

Author Response

We thank you for your comments, please find the answer point by point as follows:

  1. We thank you for your comments. We have divided the paper into the first part regarding oxidative stress and the three primary treatment data from animals and patients. So we can continue with the discussion by presenting the data regarding cardiovascular risk which is the primary purpose of the paper.
  2. We thank the reviewer, we have changed the entire text with the word “diabetic,” and we have used the words persons/subjects with diabetes
  3. We thank the reviewer we have correct the type errors throughout the text
  4. We thank the reviewer for his comment, and we have defined the acronyms
  5. We thank the reviewer for his comments, as suggested, we avoid using antidiabetic and change the word with the ones advised.
  6. We thank the reviewer for the comment. Please find in the section pharmacological treatment a paragraph with the information request regaring pioglitazone, stroke IRIS and PROactive study. We have provided also to update and add the references.
  7. We thank the reviewer for the comment. We have provided to correct the symbols, our apologies but during the upload there was an error, and we have removed the phrase "ameliorate endothelial function."
  8. We thank the reviewer for the constructive comment. As requested, we provided to change to "improve glycemic control.”
  9. We thank the reviewer for the constructive comments As requested, we provided to change the phrase
  10. We thank you for your answer. Yes, there are data both from animal studies and then in patients. Please find added the word as requested, and we thank you for your observation
  11. We thank the reviewer for the comment. As requested, we provided to change the phrase in more recent class
  12. We thank you for your comment, and we agree regarding the importance of this new treatment that will be available. Indeed we have added the terzipatide in the discussion and not in the first part because there are few data and studies regarding cardiovascular risk is still ongoing.
  13. We thank the reviewer for the observation, and we have added the drug names as requested in table 2.
  14. We thank the reviewer for your comments. As requested, we provided to change the image by adding the pancreas and not anymore the candy wrapping.
  15. We thank the reviewer for the constructive comments. As requested, we provided to update on the reference, please find the reference underlying in yellow

Reviewer 3 Report

Authors consider the studies that recent pharmacological options in type 2 diabetes and synergic mechanism in cardiovascular disease.

This article is really well written. But authors should consider abbreviations at first appearance.

The review is well structured, but certain issues are not addressed, and important refs are neglected.

Some parts are too simplified and others are not comprehensive. Many statements are not referenced. Some sentences are trivial.

Controversy in the literature should be emphasized.

Author Response

We thank you for your comment and appreciate your advice. We have fixed our issues with the paper and updated the references.